# Impact of the First Year of the COVID-19 Pandemic on Pediatric Emergency Department Attendance in a Tertiary Center in South Italy: An Interrupted Time-Series Analysis

**DOI:** 10.3390/healthcare11111638

**Published:** 2023-06-02

**Authors:** Alessandra Alongi, Francesca D’Aiuto, Cristina Montomoli, Paola Borrelli

**Affiliations:** 1Pediatric Emergency Unit, Di Cristina Hospital, Azienda Ospedaliera di Rilievo Nazionale e di Alta Specializzazione Civico Di Cristina e Benfratelli, 90127 Palermo, Italy; 2Unit of Biostatistics and Clinical Epidemiology, Department of Public Health, Experimental and Forensic Medicine, University of Pavia, 27100 Pavia, Italy; 3Laboratory of Biostatistics, Department of Medical, Oral and Biotechnological Sciences, G. d’Annunzio University of Chieti-Pescara, 66100 Chieti, Italy

**Keywords:** COVID-19, lockdown, pediatric, emergency medicine, transmissible infectious diseases, trauma, mental health, critical illness, hospitalization, interrupted time-series analysis

## Abstract

Background: The evidence shows a reduction in pediatric emergency department (PED) flows during the early stages of the COVID-19 pandemic. Using interrupted time-series analysis, we evaluated the impact of different stages of the pandemic response on overall and cause-specific PED attendance at a tertiary hospital in south Italy. Our methods included evaluations of total visits, hospitalizations, accesses for critical illnesses and four etiological categories (transmissible and non-transmissible infectious diseases, trauma and mental-health) during March–December 2020, which were compared with analogous intervals from 2016 to 2019; the pandemic period was divided into three segments: the “first lockdown” (FL, 9 March–3 May), the “post-lockdown” (PL, 4 May–6 November) and the “second lockdown” (SL, 7 November–31 December). Our results showed that attendance dropped by a mean of 50.09% during the pandemic stages, while hospitalizations increased. Critical illnesses decreased during FL (incidence rate ratio -IRR- 0.37, 95% CI 0.13, 0.88) e SL (IRR 0.09, 95% CI 0.01, 0.74) and transmissible disease related visits reduced more markedly and persistently (FL: IRR 0.18, 95% CI 0.14, 0.24; PL: IRR 0.20, 95% CI 0.13, 0.31, SL: IRR 0.17, 95% CI 0.10, 0.29). Non-infectious diseases returned to pre-COVID-19 pandemic levels by PL. We concluded that that the results highlight the specific effect of the late 2020 containment measures on transmissible infectious diseases and their burden on pediatric emergency resources. This evidence can inform resource allocation and interventions to mitigate the impact of infectious diseases on pediatric populations and the health-care system.

## 1. Introduction

The COVID-19 pandemic determined marked changes in the utilization of emergency health-care services worldwide [1]. During the early-phase pandemic response, substantial decreases in pediatric emergency department (PED) attendance rates were reported in many countries [2,3,4] as a consequence of the stringent lockdown measures adopted to reduce the spread of COVID-19, measures which were later relaxed and then partially resumed in the last months of 2020. Parents’ avoidance behaviors due to fear of nosocomial COVID-19 infection [5], along with a reduction in common causes of pediatric morbidity due to infection control measures and movement restrictions [6], were seen as a principal explanation for the changes in emergency care seeking patterns.

While multiple studies revealed a reduction mainly in low acuity and inappropriate accesses [7,8,9], mixed reports are available on critical illness presentations, with some authors reporting a drop in high acuity cases [10,11] and others describing an increase in acuity and admission rates [12,13], raising concerns due to the supposedly delayed presentation of children with serious illnesses [14]. To further add complexity, different patterns of pediatric emergency department utilization have been described for specific causes of access; while a growing body of evidence demonstrates the decreasing effect of the early pandemic response on infectious disease visit volumes [15,16,17], data on non-communicable causes of morbidity such as injuries and psychosocial issues are more heterogeneous. Some studies found a reduction in mild and severe trauma and fracture rates [18,19], with differences across age groups [20]. However, increases in specific injury patterns [21,22] as well as stable [23,24] and even raised rates of trauma-related visits [25] have also been reported. The effects of the pandemic on mental health emergency visits also appear to be complex, with accumulating evidence of an initial decrease during the first stage of public health response followed by a long-term increasing trend [26,27,28]; however, it is unclear whether these patterns are related to actual changes in pediatric populations’ morbidity or simply reflect trends in overall health-care services usage and how they interact with PED attendance for other causes [29].

Together, these findings depict a multifaceted picture of the impact of containment measures on PED attendance, suggesting potentially differential effects according to patients’ characteristics and conditions, which may fluctuate depending on the extent of public health interventions. However, most of the existing studies are limited to the first months of the pandemic, preventing the evaluation of the varying effects of the evolving pandemic containment strategies and changes in population behavior during 2020.

To address this, in this study we aimed to quantify the differential impact of the stages of the COVID-19 pandemic during 2020 and the related public health interventions on total and cause-specific pediatric emergency department (PED) visits at a tertiary hospital in south Italy.

## 2. Materials and Methods

### 2.1. Study Design, Data Source and Setting

We adopted a quasi-experimental design to evaluate the impact of three different pandemic stages in 2020 on PED attendance using a single-center interrupted time series with segmented regression analysis.

Data were gathered from the routine electronic clinical records of the emergency department of the Di Cristina Hospital (Azienda Ospedaliera ad Alta Specializzazione “Ospedali Civico Di Cristina Benfratelli”) in Palermo, Sicily, Italy. The center, a tertiary referral hospital with a pediatric intensive care unit performing around 35,000 ED visits per year, is the only pediatric hospital in Palermo and a hub serving the Sicilian region for pediatric neuropsychiatric diseases.

We extracted all the de-identified individual records of PED attendances and inpatient admissions among children aged 0–18 years in a 5-year period from January 2016 to December 2020. The records contained information about patient demographics, admission dates, main complaints, underlying medical conditions (including genetic, cardiological, neurological, nephrological, gastroenterological and oncological diseases), triage scoring, diagnosis established after the PED assessment—coded using the International Statistical Classification of Diseases and Related Health Problems, Tenth Revision (ICD-10)—and outcomes of the visit. According to the Italian triage scoring system, which uses a five-level color priority coding, non-delayable urgencies/emergencies needing immediate care were identified as “red codes” and classified as “critical illness visits” for the aim of the analysis, semi-urgent cases were identified as “yellow codes”, low-acuity visits as “green codes” and non-urgent visits as “white codes”.

### 2.2. Period Definitions

We identified the period from 1 January 2016 to 8 March 2020 as the pre-pandemic period and that from 9 March 2020 to 31 December 2020 as the pandemic period; we further defined 3 pandemic time segments a priori, based on the stages of the pandemic and the related Italian public health measures during 2020.

The Italian Government imposed a generalized lockdown from 9 March 2020; the measures adopted during this period included mobility restriction (except for work- or health-related reasons), a universal mask mandate for both indoors and outdoors, suspension of in-person school, university activities and cultural events and closures of non-essential businesses [30]. Given the decreased incidence of COVID-19 infections, public health measures have been progressively relaxed since 3 May 2020, when visits to family members, within-country travel and reopening of commercial activities were allowed; meanwhile, mask mandates and bans on large-scale meetings were kept in place [31]. A second large wave of COVID-19 in the autumn of 2020 [32] led to the implementation of a second partial lockdown from November 2020 to the end of December 2020 [33]. From 14 September 2020 to 24 September 2020, schools were reopened in Italy [34]. Starting 6 November, Italy adopted different non-pharmaceutical interventions (NPIs) at the regional level according to a color scheme (“yellow”, “orange” and “red”) corresponding to increasing levels of restrictions based on the combination of quantitative indicators of COVID-19 transmission and the estimated regional health-care resilience; during this period, Sicily was mostly assigned to an intermediate (“orange”) level of restrictions; these late 2020 containment measures encompassed a curfew between 10 pm and 5 am (as opposed to the full-day stay-home mandate in the first lockdown), closures of restaurants and bars except for takeaway service, capacity reduction for public transport and alternate distance learning in high schools (for children aged >12 years) and universities (rather than full school closures for students of all age, as in the early pandemic phase) [35]; the obligation to wear masks indoors was retained.

In this study, we therefore further divided the pandemic period (from March to December 2020) into three segments: the “first lockdown” (FL), from 9 March 2020 to 3 May 2020 (8 weeks); the “post-lockdown” (PL), from 4 May 2020 to 6 November 2020 (27 weeks) and the “second lockdown” (SL), from 7 November 2020 to 31 December 2020 (7 weeks).

### 2.3. Outcomes

The primary outcome was defined as the number of unscheduled pediatric emergency department visits per week. Elective hospital admissions for a planned procedure or treatment and revisits scheduled to complete diagnostic workups were excluded from the analysis. Secondary outcomes included the number of critical illness visits (categorized as “red code” by the Italian triage system) and the hospitalization rate. The analyses were further stratified by age groups at the time of the visit (0–6 years, 7–12 years and >12 years).

We also examined cause-specific weekly visits within separate models. Four groups of diagnoses were identified based on the assigned International Classification of Disease Version 10 (ICD-10) code at discharge as (1) transmissible infectious diseases (e.g., respiratory or gastrointestinal infections), (2) non-transmissible infectious diseases (e.g., UTIs, soft tissue and bone infections, etc.), (3) trauma-related PED visits and (4) mental health-related PED visits. The ICD-10 codes used to identify each etiological category are detailed in Appendix A.

### 2.4. Statistical Analysis

Data recorded between March and December in the years from 2016 to 2019 (pre-pandemic period) were aggregated and compared with those recorded between March and December 2020 (pandemic period).

Descriptive statistics in terms of frequency (n) and percentages (%) were calculated for categorical variables.

The normality distribution of quantitative variables was assessed using Shapiro–Wilk tests and graphical methods; accordingly, all variables that were normally distributed have been presented as mean (SD), whereas non-normally distributed variables have been summarized as median and interquartile range (IQR). The homogeneity of variances was tested with Levene’s test.

Outcomes and demographic characteristics were compared with either Pearson’s χ2 test or Fisher’s exact test for qualitative variables and Student’s *t*-test for independent data and/or the non-parametric analogous (Wilcoxon test) for the quantitative ones.

We then performed an interrupted time-series analysis using segmented (piecewise) regression to estimate the effects of the pandemic stages on the outcomes of interest.

This approach allows estimating both immediate (as changes in level) and sustained effects (increase or decrease in the slope) attributable to an intervention, while accounting for pre-intervention secular trends.

We fitted a set of generalized linear and additive models; the main features of each model are reported in Appendix B, Table A1.

Each model included the weekly count of the observed outcome as response variable (Y), a covariate for time (captured by the number of weeks from the start of the observations) and a term for step change represented by a dummy categorical variable indicating the study periods (first lockdown, post-lockdown and second lockdown, with “pre-pandemic” as the reference category); the trend for each period was operationalized using a progressive numeric variable centered on the starting week of the respective period, according to the following structure:Yi = β0 + T + β1X1 + β2X2T + β3X3T + β4X4T + ε,(1)

The equation includes Y and the following dependent variables: T, number of weeks starting in January 2016; β0, the intercept expressing the baseline level of the outcome during the pre-pandemic period; β1, the coefficient of the dummy variable representing the study periods, estimating the level change during each period; β2, β3 and β4 which estimate the trends of the time-series during the aforementioned periods and e: the error term.

For the hospitalization rate, the total number of visits per week was used as an offset.

We modeled critical illness visits using a Poisson generalized linear model; for mental-health visits, a zero-inflated Poisson model (ZIP) was adopted with a single zero-inflation parameter applied to all observations and for all the other outcomes, we applied negative binomial models to account for the detected overdispersion.

To account for nonlinear time trend and seasonality, total visits, hospitalization rate, trauma and transmissible infectious diseases counts were modeled as generalized additive (mixed) models (GAMMs), where spline functions for weeks or months and for time were included (Appendix B, Table A1).

For total accesses, hospitalization rate and transmissible infectious diseases we added a first-order autoregressive component (AR1) with a random intercept for month to control for autocorrelation.

Analyses of total visits, critical illness visits and hospitalization rates were further stratified by age groups.

To assess the differences for each outcome between pandemic periods, pairwise post hoc comparisons of estimated marginal means were carried out using Turkey’s adjustment for multiple comparisons with the emmeans package [36].

Models were selected based on virtue of fit/deviance and the LR test for nested Poisson or negative binomial models and using information criteria such as AIC for non-nested models.

Residual diagnostics plotted from the ‘simulateResiduals’ function in the DHARMa package [37] were used to examine the model assumptions of each model.

The autocorrelation Function (ACF) and the partial autocorrelation function (PACF) plots were examined for discernible patterns and autocorrelation.

The statistical significance level was set at α = 0.05.

Analyses were performed using R Software analysis, version 4.1.3, R Foundation for Statistical Computing (Vienna, Austria).

## 3. Results

From 9 March through 31 December 2020 (pandemic period), a total of 13,180 visits were registered at our facility. We observed an average decrease of 50.09% (95%CL −50.8, −51.1) in the number of accesses per week between the pre-pandemic (median 668, IQR 620–722) and pandemic (median 337, IQR 279–389) periods. The distribution of patients’ age at visit rose slightly in the pandemic period, but it remained comparable. The proportion of patients with underlying clinical conditions was lower among those who presented to the PED during the pandemic period (0.9% vs. 1.4%). During the pandemic months a relative increase in the proportion of hospitalizations was observed with 29% of total visits resulting in hospital admission, compared to 20% in the pre-pandemic period. Low acuity (green) triage codes decreased from 73% to 70%, while the percentage of semi-urgent (yellow code) visits increased from 25 to 29%; accordingly, the proportion of critical illness visits raised from 0.6% to 0.9% of the total volume, with a similar increase in intensive care unit (ICU) admissions. Among the examined causes of visits, transmissible infectious diseases constituted the main reasons of attendance in the pre-pandemic period (35% of all visits); in the pandemic period, trauma became the most frequent presentation (24%). A comparison of outcomes and demographics between the pre- and post-pandemic visits is reported in Table 1.

To further characterize patients presenting with critical illness, we carried out a comparison of demographic and clinical features and of the outcomes of red-code visits across the pre-pandemic and pandemic periods, as reported in Table 2; no difference in age, presence of comorbidities, proportion of proposed admissions, refusal of admission to wards or brief intensive observation area by parents, admissions to ICU or deaths was evidenced between the two periods. During the pandemic period we observed a relative decrease in critical illness visits associated with infectious diseases (20% vs. 29.3%) and trauma (9.7% vs. 12%), with a contextual increase in the proportion of neurological etiologies (e.g., seizures), which became the main cause of critical illness (34% vs. 25%), although differences were not significant.

### 3.1. Total Visits

The onset of the first lockdown was associated with an abrupt drop of 69% of weekly visits (IRR 0.31, 95% CI 0.27, 0.37); after the first days of lockdown, visit counts followed a growing trend, with an increase of around 7% per week during this stage (IRR 1.07, 95% CI 1.04, 1.11); a similar reduction was also observed in the post-lockdown period (IRR 0.36, 95% CI 0.27, 0.48), with an increasing trend of 8% per week (IRR 0.92, 95% CI 0.89, 0.96), and in the second lockdown (IRR 0.40, 95% CI 0.29, 0.57), also with a trend toward reduction (IRR 0.94, 95% CI 0.90, 0.98) (Table 3). In Figure 1, estimated trends for total weekly visits are compared to the counterfactual scenario assuming no effects of the pandemic stages.

Pairwise comparisons of estimated marginal means showed a significant reduction in weekly visits during each of the three stages compared to the pre-COVID period, but no significant differences across the pandemic stages (Appendix C, Table A2).

The reduction in the total visits counts was evident in all age groups throughout all study stages, showing similar increasing trends during the first lockdown and decreasing trends in the subsequent stages, although the reduction trend for patients older than 12 years was not significant (Appendix C, Table A3).

### 3.2. Critical Illness Visits

There was a decrease in critical illness visits during both the first lockdown (IRR 0.37, 95% CI 0.13, 0.88) and the second lockdown (IRR 0.09, 95% CI 0.01, 0.74), with steady trends throughout all three stages (Table 4, Figure 2); however, contrast comparisons did not show significant differences between the lockdowns and the pre-pandemic periods (Appendix C, Table A2; Appendix D, Figure A1b). Stratificated analyses by age found no significant differences in the levels and trends of critical illness volumes for specific age groups (Appendix C, Table A3).

### 3.3. Hospitalization Rate

We observed a significant increase in the hospitalization rate during the first lockdown (IRR 1.86, 95% CI 1.57, 2.21), post-lockdown (IRR 1.68, 95% CI 1.24, 2.27) and second lockdown (IRR 1.59, 95% CI 1.10, 2.29) (Figure 3). Hospitalization trends were declining during the first lockdown (IRR 0.95, 95% CI 0.92, 0.99), increasing in the post-lockdown stage (IRR 1.06, 95% CI 1.02, 1.10) and were stationary during the second lockdown (IRR 1.00, 95% CI 0.96, 1.04).

Pairwise comparisons across the study periods revealed that weekly admission rates were significantly different from the pre-pandemic period only in the first lockdown and the post-lockdown stages (Appendix C, Table A2; Appendix D, Figure A1c).

Age-stratified analyses showed that while the rise in hospitalization rate during the first lockdown was consistent across all age groups, throughout the post-lockdown and second lockdown stages the increase was evident only for patients in the 0–6 and 7–12 years groups (Appendix C, Table A3).

### 3.4. Cause-Specific Visits

Model estimates from interrupted series analyses for each etiological category are shown in Table 5 and Figure 4. Figure 5 charts trends in cause-specific PED volumes during the pre-pandemic and pandemic periods.

#### 3.4.1. Transmissible Infectious Diseases

We observed a sharp decrease in transmissible infectious diseases visits during all the pandemic stages, with a reduction of around 80% (FL: IRR 0.18, 95% CI 0.14, 0.24; PL: IRR 0.20, 95% CI 0.13, 0.31, SL: IRR 0.17, 95% CI 0.10, 0.29).

A declining trend was evident during the second lockdown (IRR 0.91, 95% CI 0.85, 0.97). Pairwise comparisons with the pre-pandemic period demonstrated significant reductions for all stages (Appendix C, Table A2; Appendix D, Figure A1d).

#### 3.4.2. Non-Transmissible Infectious Diseases

A decline in non-transmissible infectious diseases visits was apparent during the first lockdown (IRR 0.10, 95% CI 0.04, 0.25) and the second lockdown (IRR 0.23, 95% CI 0.06, 0.92); however, the only significant pairwise difference was found between the pre-pandemic period and the first lockdown (Appendix C, Table A2; Appendix D, Figure A1e).

#### 3.4.3. Trauma

There was a significant reduction in levels and time trends of trauma-related accesses during the first lockdown (IRR 0.23, 95% CI 0.06, 0.92) and post-lockdown phases (IRR 0.23, 95% CI 0.06, 0.92).

Pairwise comparisons indicated a significant reduction between pre- and first lockdown, but not post-lockdown levels; moreover, an upsurge of trauma-related visits was evident during the second lockdown compared to the post-lockdown stage (estimated marginal means 108 vs. 70, *p* < 0.001), indicating a return of injury visit volumes to pre-pandemic levels in this segment (Appendix C, Table A2; Appendix D, Figure A1f).

#### 3.4.4. Mental Health

A reduction in mental health-related visits (IRR 0.23, 95% CI 0.06, 0.92) was associated with the first lockdown; however, pairwise contrasts did not reveal significant differences (Appendix C, Table A2; Appendix D, Figure A1g).

## 4. Discussion

We conducted a retrospective 5-year chart review to evaluate the differential effects of the stages of the COVID-19 containment response on total and cause-specific PED attendance during the first year of the pandemic in a tertiary referral center in south Italy.

While a wide body of evidence established the disruption of health-care systems during the first months of strict measures, the longer-term impact of COVID-19 on pediatric emergency services utilization, including the later phases of the pandemic containment when restrictions were partially relaxed, has been less investigated.

In line with previous evidence [9,38], we observed a sharp decrease in the total volume of visits during the first two months of the pandemic for all age groups and causes of access, including for patients with critical illness, with a simultaneous increase in acuity and hospitalization rates; during the final stage of the 2020 pandemic response, however, this effect appeared specifically linked to a reduction in transmissible infectious disease related visits.

Assessing the impact of COVID-19 policies on health-care utilization presents several methodological challenges due to the confounding effects of a variety of epidemiological factors and interventions [39].

We adopted interrupted time-series analysis, one of the strongest quasi-experimental designs, which carries several advantages compared to more basic (pre/post) observational design as it allows and adjusts for short-term fluctuations, pre-existing underlying patterns, nonlinear trends and seasonality that are known to be especially important for rates of infectious disease related visits [40].

Although the effects of COVID-19 lockdowns on pediatric health-care utilization have been extensively examined worldwide, relatively few studies have examined time series of PED visits with this approach [41].

Health-care avoidance by patients and parents due to fear of COVID-19 exposure has been cited by several authors as the main factor explaining the reduction in emergency care burden reported worldwide [42,43].

Unmet health-care needs, due to real or perceived access barriers during the pandemic, also raised concerns as reduced or delayed access to medical care has been linked to worse outcomes in patients with severe diseases, particularly in high-risk populations [44,45].

As children with underlying clinical conditions are expected to suffer a worse outcome from (hospital-acquired) infections, health-care avoidance could contribute to an explanation of the lower percentage of patients with comorbidities presenting to PED during the pandemic periods in our data.

Overall, however, we found no evidence of a worse outcome for patients presenting with critical illness during the pandemic phase due to delayed presentation or care seeking avoidance, as admissions to ICUs and deaths among these patients were comparable between the study periods.

Several groups observed that the reduced visit volume was attributable mainly to a decrease in the number of non-urgent visits, leading to a proportional increase in acuity and hospitalization rates [2,7,8,20].

Our findings confirm this trend towards a rise in hospital admissions, which was persistent during all stages of the 2020 pandemic, although in the last two pandemic stages it was evident only for children under 12 years of age.

It has been proposed [20] a possible explanation could lie in the increased availability of hospital beds itself, which is known to correlate positively with inpatient admissions due to the so-called Roemer effect [46].

Accordingly, we found a reduction in low acuity accesses (green codes, as defined by the Italian triaging system) during the pandemic period.

Thus, as already highlighted, public health measures could have caused a reduction in inappropriate health-care system use and low-value care [1,47].

Nevertheless, we also found a tendency towards a reduction in critical illness visits, especially during the first lockdown; this has been relatively rarely signaled in the literature [10], in contrast with the results of other studies [8,23].

While distinguishing between the effects of health-care avoidance and those of public health interventions on the prevalence of many pediatric diseases remains challenging, it has been observed [10,48] that infectious diseases represent a major cause of critical illness in pediatric patients (i.e., febrile seizures, septic shock, respiratory distress, etc.), to an extent that could explain the effect of the reductions in pathogen transmission due to pandemic mitigation measures on critical patient visit volumes.

We observed an absolute and relative decrease of ‘red codes’ associated with infectious diseases during the pandemic period, suggesting that the reduction in pathogens circulation contributed to lower critical illness visit volumes.

Single, multicentric and population-wide studies have reported large reductions in respiratory pathogen circulation and related hospitalization in children [49,50,51].

Moreover, many authors reported significantly greater reductions in communicable infections compared to non-infectious diagnoses [4] or non-transmissible infections [16,49].

Consistent with other findings worldwide [50,52,53], we found that the drop in transmissible infectious diseases was the most substantial among the examined causes of access and persisted, albeit to a lesser extent, after the first lockdown throughout the last months of 2020.

Crucially, during this stage, our analysis evidenced a significant decrease specifically in transmissible infectious disease related visits, mainly led by viral infections, while other major causes of PED access, such as traumas and non-transmissible infectious diseases, were stable or returned to pre-pandemic levels.

In November and December 2020, public health measures to contain COVID-19 in Italy included alternate distance schooling for older children, face mask mandates and partial restrictions of social gatherings.

Notably, an Italian study on PED attendance during school and commercial activities reopening in October 2020 revealed a deeper decrease in respiratory infectious disease rates compared to non-respiratory communicable infectious disease rates, suggesting that airborne transmission mediated the impact of preventive measures on infectious disease prevalence [54].

Interestingly, as the collective measures adopted in 2020, such as mask wearing and distance learning, were lifted in many countries, the ensuing years saw a resurgence of pediatric communicable diseases worldwide, with atypical timing trends [55,56]; in the winter of 2022, in particular, the incidence and severity of respiratory infections such as influenza A and RSV put pediatric hospital services under unprecedented pressure [57].

Taken together, these findings corroborate the potential role of partial, less restricting NPIs—such as face masking and partial distance learning—in reducing the morbidity and health-care costs of pediatric infectious diseases by their effects on the circulation of airborne pathogens.

Social and behavioral changes due to “shelter-in-place” orders, closure of schools and sport facilities and mobility restrictions had likely impacted the prevalence of other common causes of pediatric morbidity, namely traumatic injuries.

Expectedly, decreases in injury-related referrals in adult populations have been extensively reported [58].

For pediatric patients, however, some authors reported an increase in trauma-related PED visits, particularly in younger children, which has been linked to a higher risk of domestic accidents during the first months of the pandemic, when stay-at-home orders were adopted [20,59].

In our facility, trauma-related accesses showed a sizable decrease during the first two phases of the pandemic, with a relative rise during the second lockdown in which visit counts due to injuries returned to levels comparable to those registered before the beginning of the pandemic.

We also observed an overall reduction in trauma-related critical illness visits during the pandemic period, which could be accounted for by a decrease in traffic accidents associated with the mobility restrictions.

While we could not compare the etiology of reported injuries across the periods, considering that injuries such as wounds and fractures are more likely to require hospital care than infectious and medical diseases, this finding suggests that there was an effect of the decreased activities and mobility during the first lockdown that gradually receded with the relaxation of public restrictions and subsequent resumption of activities in the last months of 2020.

Other potential impacts of the COVID-19 pandemic on the morbidity of pediatric populations, which have been wildly explored in the published literature, concern mental health [60].

Early studies mainly reported a dramatic decrease in PED attendance for psychosocial issues during the first months of the pandemic [61], in line with the overall decrease in utilization of health-care services.

Later, longer time-series studies showed an increase in pediatric mental health referrals and related admissions in the second half of 2020 and beyond [62,63,64], including eating disorders [28], self-harm [65] and suicidality [63,66,67].

However, these findings may reflect the resumed access to mental health services after the first month of strict restrictions, or the restored exposures to relevant social stressors (i.e., schools reopening) [68,69], as much as an adverse effect of the public health measures on children’s mental health [66,70].

Contrary to the latter hypothesis, in our study, while mental health visits declined during the first lockdown following the general trend, we found no evidence of increases for this cause of access during the second lockdown, even in age groups (>12 years) that experienced in-person schooling restrictions.

These results must be interpreted in light of several limitations.

First, this is a single-center study whose generalizability may be limited to similar health-care contexts.

Due to the retrospective nature of the analysis, we did not carry out a formal power analysis, and we cannot exclude the risk of false negative findings due to low statistical power, especially for low prevalence causes of access.

Moreover, we could not consider contextual changes in access to other health-care facilities for the examined population in the catchment area, nor the implementation of other resources for the management of non-urgent conditions such as telemedicine or remote consultations.

Finally, we could not consider the dynamics of the spread of COVID-19 in our population and specifically the number of PED entrances due to SARS-CoV-2 infections, as it was mostly diagnosed after admission to our center.

However, given that Sicily showed a relatively low prevalence of COVID-19 [71]—especially when compared to other Italian regions and for pediatric patients—during the examined periods, we are confident that this omission would have little appreciable impact on the results.

## 5. Conclusions

Our findings add to the growing literature on the effects of public health measures on the utilization of emergency health-care services and causes of morbidity in the pediatric population.

Comparing the patterns of cause-specific PED accesses can help in understanding the sensitivity of these conditions to different grades of public health interventions.

In particular, the specific and persistent reduction we observed in 2020 for transmissible infections, compared to other causes of access, underscores the effectiveness of relatively looser NPIs during school and work activities reopening in decreasing the spread of communicable diseases in children and the related burden on hospital resources.

As known infectious pathogens resurface and new COVID-19 variants emerge, it is important to consider this evidence to develop potential strategies for preventing and mitigating their impact on populations and emergency services.

Moreover, our analysis revealed the influence of early restrictions on the ability of parents to seek potentially life-saving care for children with medical emergencies.

As health-care avoidance due to fear of exposure to infectious pathogens in hospital settings disproportionally affects patients with underlying conditions or chronic diseases, who are more at risk of developing critical illness, these findings highlight the importance of interventions aiming to minimize the spread of airborne hospital-acquired infections to ensure safe access to timely care for all patients.

These perspectives have important implications for the planning of pediatric health-care resource allocation in the next years, as well as to inform future epidemiologic research.

In the next years, the ongoing monitoring of PED utilization rates and patterns will be pivotal to ensure the appropriate reorganization of emergency care services in the face of the ever-evolving epidemiological landscape.

## Figures and Tables

**Figure 1 healthcare-11-01638-f001:**
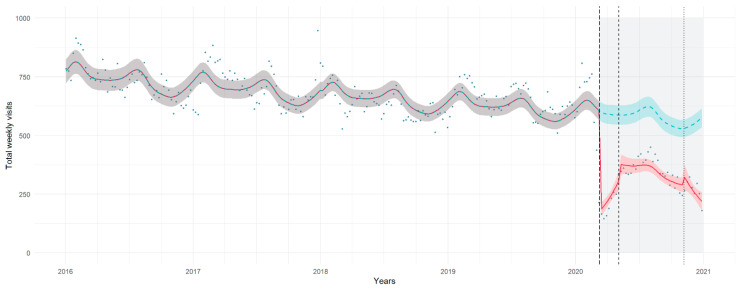
Count of total visits per week (y-axis) during the examined years (x-axis); the pandemic months are highlighted in light grey, with the start of the first lockdown, post-lockdown and second lockdown marked by vertical lines (longdashed, dashed and dotted, respectively). Observed counts are represented by dots; the chart shows the estimated trends for the study model (red dashed line) and the counterfactual scenario of the predicted number of weekly visits assuming the pre-pandemic pattern continuing in the pandemic period (green dashed line). Shadows represent 95% prediction intervals.

**Figure 2 healthcare-11-01638-f002:**
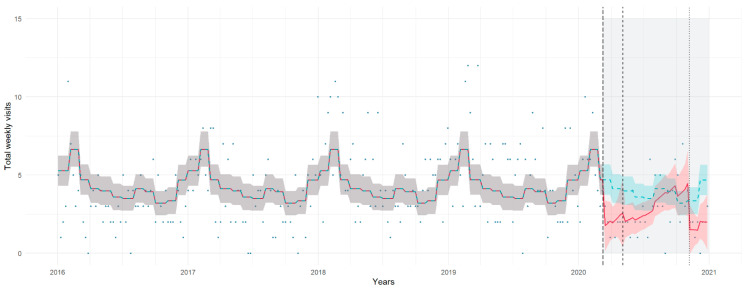
Critical illness visits per week during the pre-pandemic and pandemic periods (light gray area), with trendlines for the study model (red) and the counterfactual scenario (green). Shadows represent 95% prediction intervals.

**Figure 3 healthcare-11-01638-f003:**
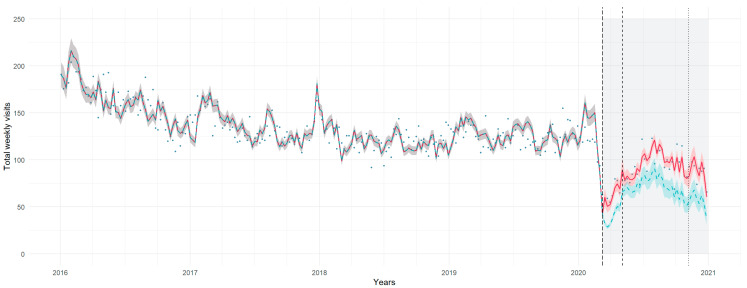
Hospitalizations per week during the pre-pandemic and pandemic periods (light gray area), with trendlines for the study model (red) and the counterfactual scenario (green). Shadows represent 95% prediction intervals.

**Figure 4 healthcare-11-01638-f004:**
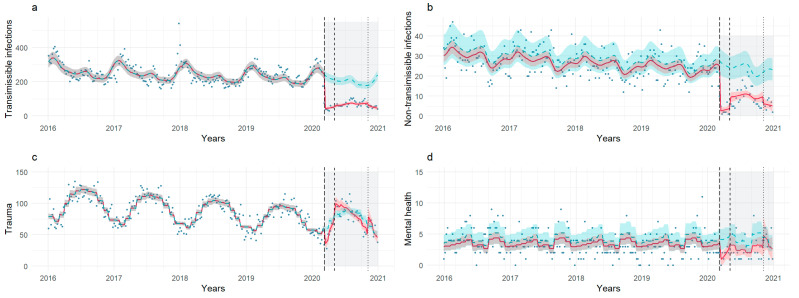
Count of weekly visits for each etiological category: transmissible infectious disease (**a**), non-transmissible infectious disease (**b**), trauma (**c**) and mental health (**d**). The start of the three pandemic stages is marked by dashed/dotted lines. The pandemic months are highlighted in light gray; trendlines for the study model (red) and the counterfactual scenario (green) are shown. Shadows represent 95% prediction intervals.

**Figure 5 healthcare-11-01638-f005:**
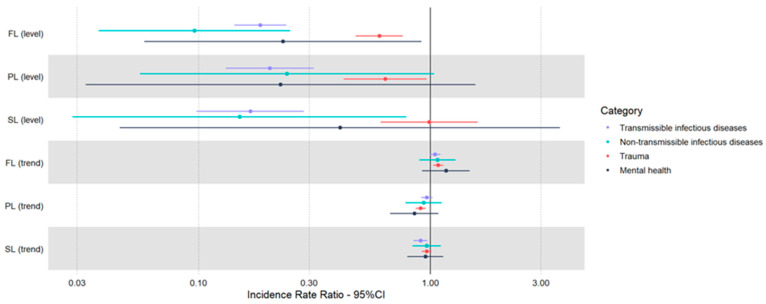
Forest plot of incidence rate ratios (IRR) from an interrupted time-series models expressing changes in level and trend during pandemic stages for each etiological category. FL: first lockdown; PL: post-lockdown and SL: second lockdown.

**Table 1 healthcare-11-01638-t001:** Comparison of outcomes and demographic characteristics across the pre-pandemic and pandemic periods. Data are aggregated by week and expressed as numbers (percentages) unless otherwise specified.

Variable	Pandemic Period, N = 13,180	Pre-Pandemic Period, N = 112,701	*p*-Value
Total visits	337 (279–389) ^1^	668 (620–722) ^1^	<0.001 ^2^
Triage code			<0.001 ^3^
Non urgent (white code)	101 (0.8%)	881 (0.8%)	
Low acuity (green code)	9201 (70%)	78,231 (73%)	
High acuity (yellow code)	3765 (29%)	27,020 (25%)	
Dead on arrival (black code)	0 (0%)	23 (<0.1%)	
Critical illness visits (red code)	113 (0.9%)	649 (0.6%)	<0.001 ^4^
Hospitalizations	3769 (29%)	22,502 (20%)	<0.001 ^4^
Transmissible infectious diseases	2759 (21%)	39,008 (35%)	<0.001 ^4^
Non-transmissible infectious diseases	330 (2.5%)	4425 (3.9%)	<0.001 ^4^
Trauma	3112 (24%)	15,961 (14%)	<0.001 ^4^
Mental health	115 (0.9%)	610 (0.5%)	<0.001 ^4^
ICU admissions	33 (0.3%)	168 (0.1%)	0.006 ^4^
Age (yrs.)	5.0 (2.0–10.0) ^1^	5.0 (2.0–9.0) ^1^	<0.001 ^1^
Age (strata)			<0.001 ^4^
0–6	7583 (58%)	67,573 (60%)	
7–12	4150 (31%)	34,441 (31%)	
>12	1447 (11%)	10,687 (9.5%)	
Comorbidity	122 (0.9%)	1528 (1.4%)	<0.001 ^3^

^1^ Median (IQR); ^2^ Wilcoxon’s rank test; ^3^ Fisher’s exact test; ^4^ Pearson’s chi-squared.

**Table 2 healthcare-11-01638-t002:** Comparison of clinical-demographic characteristics and outcomes of patients presenting with critical illness in the pre-pandemic and pandemic periods. Data are expressed as numbers (percentages) unless otherwise specified.

Characteristic	Pandemic Period, N = 113	Pre-Pandemic Period, N = 647	*p*-Value
Age	4.0 (2.0–8.0) ^1^	4.0 (2.0–8.0) ^1^	0.600 ^2^
Comorbidity	1 (0.9%)	18 (2.8%)	0.300 ^3^
*Visit outcome*			
Proposed admission	93 (82%)	512 (79%)	0.500 ^3^
Refused admission	5 (4.4%)	23 (3.6%)	0.600 ^3^
Refused brief intensive observation	6 (5.3%)	17 (2.6%)	0.140 ^3^
Deceased	1 (0.9%)	2 (0.3%)	0.400 ^3^
ICU admission	13 (12%)	75 (12%)	0.999 ^3^
*Etiology*			
Transmissible infectious diseases	23 (20%)	188 (29%)	0.068 ^3^
Non-transmissible infectious diseases	0 (0%)	2 (0.3%)	0.999 ^3^
Trauma	11 (9.7%)	76 (12%)	0.600 ^3^
Mental health	0 (0%)	3 (0.5%)	0.999 ^3^
Neurologic diseases	38 (34%)	163 (25%)	0.065 ^3^

^1^ Median (IQR); ^2^ Wilcoxon’s rank test; ^3^ Fisher’s exact test.

**Table 3 healthcare-11-01638-t003:** Interrupted series analysis results for total visits. Incidence rate ratios (IRR) express changes in level and trend during different pandemic stages. The model included a spline function of week and time. FL: first lockdown; PL: post-lockdown and SL: second lockdown.

Variable	IRR	95% CI	*p*-Value
**Level**			
FL	0.31	0.27, 0.37	<0.001
PL	0.36	0.27, 0.48	<0.001
SL	0.40	0.29, 0.57	<0.001
**Trend**			
FL	1.07	1.04, 1.11	<0.001
PL	0.92	0.89, 0.96	<0.001
SL	0.94	0.90, 0.98	0.006
s (week)			<0.001
s (time)			<0.001

**Table 4 healthcare-11-01638-t004:** Interrupted series analysis results for critical illness visits. Incidence rate ratios (IRR) express changes in level and trend during different pandemic stages. The model included a dummy variable indicating the month, for seasonal adjustment. FL: first lockdown; PL: post-lockdown and SL: second lockdown.

Variable	IRR	95% CI	*p*-Value
**Level**			
FL	0.37	0.13, 0.88	0.040
PL	0.28	0.05, 1.62	0.200
SL	0.09	0.01, 0.74	0.027
**Trend**			
FL	1.08	0.87, 1.34	0.500
PL	0.96	0.77, 1.19	0.700
SL	0.95	0.75, 1.22	0.700
**Month**			
January	—	—	—
February	1.26	0.98, 1.62	0.068
March	0.89	0.68, 1.17	0.400
April	0.78	0.59, 1.04	0.094
May	0.76	0.56, 1.01	0.062
June	0.68	0.50, 0.92	0.012
July	0.67	0.50, 0.89	0.006
August	0.78	0.59, 1.04	0.093
September	0.75	0.56, 0.99	0.042
October	0.61	0.45, 0.82	0.001
November	0.65	0.47, 0.88	0.006
December	0.90	0.68, 1.18	0.400

**Table 5 healthcare-11-01638-t005:** Interrupted series analysis results for cause-specific visits. Incidence rate ratios (IRR) express changes in level and trend during different pandemic stages. FL: first lockdown; PL: post-lockdown and SL: second lockdown.

	Transmissible Infectious Disease			Non-Transmissible Infectious Disease			Trauma			Mental Health		
Level	IRR	95% CI	*p*	IRR	95% CI	*p*	IRR	95% CI	*p*	IRR	95% CI	*p*
FL	0.18	0.14, 0.24	<0.001	0.1	0.04, 0.25	<0.001	0.6	0.48, 0.76	<0.001	0.23	0.06, 0.92	0.037
PL	0.2	0.13, 0.31	<0.001	0.24	0.06, 1.04	0.056	0.64	0.42, 0.96	0.033	0.23	0.03, 1.57	0.130
SL	0.17	0.10, 0.29	<0.001	0.15	0.03, 0.79	0.025	0.99	0.61, 1.60	>0.9	0.41	0.05, 3.63	0.400
**Trend**												
FL	1.05	0.99, 1.11	0.081	1.07	0.90, 1.28	0.400	1.08	1.03, 1.14	0.002	1.17	0.92, 1.48	0.200
PL	0.97	0.91, 1.02	0.200	0.94	0.78, 1.12	0.500	0.91	0.86, 0.96	<0.001	0.85	0.67, 1.08	0.200
SL	0.91	0.85, 0.97	0.006	0.97	0.84, 1.11	0.600	0.96	0.92, 1.01	0.150	0.95	0.80, 1.14	0.600

## Data Availability

The data presented in this study are available on request from the corresponding author. The data are not publicly available due to data protection issues.

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
