# Peer review of "Impact of the First Year of the COVID-19 Pandemic on Pediatric Emergency Department Attendance in a Tertiary Center in South Italy: An Interrupted Time-Series Analysis"

_healthcare, 2023, doi:10.3390/healthcare11111638_

Round 1
Reviewer 1 Report
Dear authors,
I read with interest your article on PED during the pandemic and I congratulate you to the precise and conclusive analysis you present in the manuscript.
I have some remarks:
1) IRR in the abstract should be explained.
2) As red code admissions increased slightly it would be very interesting to get informations on the outcome of these patients. These data would give an idea if some patients presented later to the ED in the disease course during the pandemic.
3) The results section would improve if some of the data (e.g. tab. 3) would be moved to the appendix.
4) Did I understand it right that hospitalisations decreased but hospitalisation rate increased? Maybe you could state this clearer in the results section.
Reviewer 2 Report
Overall, this manuscript presents an interesting study on the impact of COVID-19 pandemic response on pediatric emergency department (PED) attendance and hospitalizations at a tertiary hospital in South Italy. The use of interrupted time series analysis to evaluate the different stages of the pandemic response adds a valuable perspective to the literature on this topic. The comparison of PED attendance during the pandemic period with analogous intervals from 2016 to 2019 is appropriate, but it would be helpful to include more information on the characteristics of the patient population during these periods. Were there any differences in the age distribution, gender, or underlying medical conditions of patients between the pandemic and pre-pandemic periods that could have affected the results? The authors mention that critical illnesses decreased during the "first lockdown" and "second lockdown," but it would be useful to provide more details on what types of critical illnesses were included in this analysis. Additionally, it would be helpful to discuss whether the decrease in critical illnesses was due to fewer patients seeking care for these conditions or if there were other factors contributing to this trend. Furthermore, the manuscript highlights the specific impact of the late 2020 containment measures on transmissible infectious diseases. However, it would be helpful to provide more context on the nature of these containment measures and how they differed from earlier measures implemented during the pandemic. Overall, this manuscript provides important insights into the impact of COVID-19 pandemic response on PED attendance and hospitalizations, particularly regarding transmissible infectious diseases. With some minor revisions and additional information, this manuscript can be considered for publication.
